# Adaptive Normalization Enhances the Generalization of Deep Learning Model in Chest X-Ray Classification

**DOI:** 10.3390/jimaging12010014

**Published:** 2025-12-28

**Authors:** Jatsada Singthongchai, Tanachapong Wangkhamhan

**Affiliations:** Department of Computer Science and Information Technology, Faculty of Science and Health Technology, Kalasin University, Kalasin 46000, Thailand; jatsada.si@ksu.ac.th

**Keywords:** adaptive preprocessing, histogram standardization, percentile-based ROI cropping, chest X-ray classification, domain shift, cross-dataset generalization

## Abstract

This study presents a controlled benchmarking analysis of min–max scaling, Z-score normalization, and an adaptive preprocessing pipeline that combines percentile-based ROI cropping with histogram standardization. The evaluation was conducted across four public chest X-ray (CXR) datasets and three convolutional neural network architectures under controlled experimental settings. The adaptive pipeline generally improved accuracy, F1-score, and training stability on datasets with relatively stable contrast characteristics while yielding limited gains on MIMIC-CXR due to strong acquisition heterogeneity. Ablation experiments showed that histogram standardization provided the primary performance contribution, with ROI cropping offering complementary benefits, and the full pipeline achieving the best overall performance. The computational overhead of the adaptive preprocessing was minimal (+6.3% training-time cost; 5.2 ms per batch). Friedman–Nemenyi and Wilcoxon signed-rank tests confirmed that the observed improvements were statistically significant across most dataset–model configurations. Overall, adaptive normalization is positioned not as a novel algorithmic contribution, but as a practical preprocessing design choice that can enhance cross-dataset robustness and reliability in chest X-ray classification workflows.

## 1. Introduction

Chest radiography (CXR) is a widely used imaging modality due to its rapid acquisition, low cost, and non-invasive nature, supporting diagnosis across a broad range of thoracic diseases [1,2]. With advances in artificial intelligence (AI), deep learning-based CXR analysis has shown promise in improving triage efficiency and diagnostic support, particularly in high-volume and resource-limited clinical settings [1,3,4].

Despite these advances, the generalization of deep learning (DL) models across institutions remains limited. Two closely related factors contribute to this challenge. First, resizing high-resolution CXRs to standard network input sizes (e.g., 256 × 256 pixels) reduces the relative lung-to-image ratio, allowing non-diagnostic background structures to dominate learned representations and obscure subtle abnormalities [5,6]. Second, substantial domain shift arises from heterogeneity in imaging devices, acquisition protocols, contrast, brightness, and preprocessing pipelines across institutions, which can significantly degrade model performance when deployed beyond the training domain [7,8].

Image normalization is a central yet under-benchmarked factor in addressing these issues. While conventional strategies such as min–max scaling and Z-score normalization are widely adopted, their effectiveness under cross-dataset variability has rarely been evaluated in a controlled and systematic manner. Prior studies have explored adaptive instance normalization [9], contrastive domain alignment [10,11,12], multimodal integration with clinical metadata [2], and histogram-based or localized intensity normalization techniques [13,14,15,16]. However, these approaches are typically evaluated within single datasets or fixed model architectures, making it difficult to isolate the role of normalization from confounding factors such as dataset scale, model capacity, or training protocol.

To address this gap, this study conducted a systematic benchmarking analysis of three normalization strategies—min–max scaling, Z-score normalization, and an adaptive preprocessing pipeline combining percentile-based ROI cropping with histogram standardization. The evaluation was performed across four public CXR datasets, including CheXpert [17], MIMIC-CXR [18], and ChestX-ray14 [6], using three convolutional neural network architectures: a lightweight CNN, EfficientNet-B0, and MobileNetV2, under controlled sampling and identical training conditions.

This work is positioned as a controlled benchmarking study rather than a methodological innovation. The main contributions are threefold:(1)It establishes a controlled cross-dataset and cross-architecture evaluation framework for comparing normalization strategies;(2)It quantifies the impact of normalization choices on cross-domain generalization, training stability, and performance consistency, with particular emphasis on lightweight architectures such as MobileNetV2; and(3)It provides a statistically grounded comparison using Friedman–Nemenyi and Wilcoxon signed-rank tests to clarify when adaptive normalization yields meaningful performance gains over conventional approaches.

The remainder of this paper is organized as follows. Section 2 reviews related work on normalization and domain generalization in CXR analysis. Section 3 describes the datasets, normalization techniques, and experimental methodology. Section 4 presents the experimental results. Section 5 discusses the implications and limitations of the findings, and Section 6 concludes the study and outlines directions for future work.

## 2. Background and Related Work

Robust evaluation of normalization strategies for chest X-ray (CXR) classification requires datasets that reflect meaningful diversity in patient populations, acquisition protocols, and imaging conditions. Prior studies have demonstrated that deep learning models for CXR are highly sensitive to domain shifts introduced by heterogeneity in imaging sources and preprocessing pipelines [7,8,19]. Accordingly, the datasets and preprocessing techniques reviewed in this section provide the foundation for cross-dataset benchmarking of normalization methods.

### 2.1. Datasets

#### 2.1.1. ChestX-ray14

ChestX-ray14 contains over 112,000 frontal CXR images from more than 30,000 patients, labeled with fourteen thoracic disease categories [6,20]. Due to label noise arising from automated report mining, it is commonly used to evaluate model robustness under imperfect supervision [8,21].

#### 2.1.2. CheXpert

CheXpert includes more than 220,000 images from approximately 65,000 patients and explicitly models diagnostic uncertainty in its labels [17]. This design supports the evaluation of calibration, robustness, and uncertainty-aware learning in CXR classification [15,22].

#### 2.1.3. MIMIC-CXR

MIMIC-CXR comprises over 370,000 CXR images collected across multiple departments and imaging devices, introducing substantial acquisition heterogeneity [18]. This diversity makes it a key benchmark for assessing cross-domain generalization in realistic clinical settings [23].

#### 2.1.4. Pediatric Chest X-Ray (Kermany Dataset)

The pediatric CXR dataset contains 5863 images labeled as normal, bacterial pneumonia, or viral pneumonia [24]. Despite its smaller scale, it represents a distinct anatomical domain and is frequently used to study transfer learning and adult-to-pediatric generalization [25,26].

### 2.2. Preprocessing Techniques

Preprocessing is essential for mitigating domain shift caused by variations in anatomy, acquisition parameters, and institutional imaging protocols [7,11,27]. Among preprocessing strategies, normalization plays a central role in stabilizing intensity distributions, while data augmentation provides complementary benefits through geometric and intensity transformations [28].

#### 2.2.1. Normalization

Normalization aims to reduce intensity variability induced by scanner differences, exposure settings, and patient conditions. Conventional techniques such as min–max scaling and Z-score normalization remain widely used due to their simplicity and compatibility with convolutional architectures [29]. However, their effectiveness is limited in multi-source settings with heterogeneous contrast distributions.

Recent work has explored adaptive and spatially aware normalization schemes, including adaptive instance normalization and localized histogram-based methods, which demonstrate improved robustness in cross-domain and self-supervised learning pipelines [14,16,30]. These studies highlight the importance of region-focused intensity transformations for heterogeneous CXR datasets.

#### 2.2.2. Min–Max Scaling as a Baseline

Min–max scaling linearly maps pixel intensities to a fixed range and is commonly used as a baseline due to its low computational cost [31,32]. Nevertheless, it performs poorly under cross-dataset variability because it does not account for differences in underlying intensity distributions [33,34].

#### 2.2.3. Z-Score Normalization as a Standard Baseline

Z-score normalization standardizes intensities to zero mean and unit variance and has demonstrated greater resilience to device heterogeneity than min–max scaling in multi-center studies [35]. It is also associated with improved calibration and reduced performance variance in domain-adaptive frameworks [13,36].

### 2.3. Model Architectures

To assess normalization effects under realistic deployment constraints, three convolutional architectures with varying computational complexity are commonly adopted: lightweight CNNs for resource-constrained environments, EfficientNet-B0 for balanced accuracy and efficiency, and MobileNetV2 for mobile and embedded applications [37,38]. Training these architectures under identical optimization settings enables the isolation of normalization effects from architectural factors.

### 2.4. Region of Interest and Signal-to-Noise Ratio

Resizing high-resolution CXRs to standard input resolutions can reduce the lung-to-image ratio and obscure subtle pulmonary findings [1,6]. ROI-based cropping techniques mitigate this effect by preserving diagnostically relevant regions. Percentile-based cropping defined in relative coordinate space has been shown to provide stable anatomical coverage across datasets with varying resolutions and aspect ratios, supporting improved cross-dataset generalization [11].

### 2.5. Domain Adaptation and Histogram Standardization

Domain shifts caused by inter-dataset variability in brightness and contrast can significantly degrade model performance [11,18]. Histogram standardization aligns image intensity statistics to a reference distribution and has been shown to reduce inter-dataset variability, particularly when combined with region-focused preprocessing strategies [16,39].

### 2.6. Comparative Analysis of Related Work

Recent studies emphasize that normalization choices substantially influence cross-domain robustness in CXR classification [35,36]. While advances in segmentation-driven pipelines, federated learning, and model compression address complementary challenges [15,40,41], systematic comparisons of normalization strategies across datasets and architectures remain limited. This gap motivates the benchmarking framework adopted in this study.

Building on this body of work, our study presents a systematic benchmarking of normalization strategies across multiple datasets and CNN architectures, as summarized in Table 1.

### 2.7. Transformer and Foundation Model Approaches

Transformer-based and foundation model approaches, including vision–language pretraining and hybrid CNN–transformer architectures, have demonstrated strong transfer and zero-shot performance in CXR analysis [22,48,49,50,51]. Although this study focused on convolutional backbones for controlled benchmarking, the preprocessing issues examined—ROI selection, intensity normalization, and cross-domain robustness, remain directly relevant to transformer-based and foundation model pipelines, which also depend on stable input distributions for reliable downstream performance.

## 3. Methodology

### 3.1. Dataset Description

This study utilized four publicly available chest X-ray datasets that represent diverse imaging conditions, patient populations, and diagnostic labels. Three large-scale adult datasets, ChestX-ray14, CheXpert, and MIMIC-CXR, were uniformly sampled to 16,000 images each in order to formulate a controlled evaluation environment where normalization effects can be compared under identical sample sizes. The pediatric Chest-X-ray-Pneumonia dataset contains 5863 images, and the entire set was included due to its smaller size. For the pediatric dataset, class-proportion balancing was applied to mitigate label imbalance before training.

Standardizing the dataset sizes helps ensure that performance differences across normalization methods are not influenced by variations in dataset scale. This controlled sampling strategy follows recommendations from prior evaluation frameworks that emphasize reproducibility and statistical fairness when comparing preprocessing approaches in medical imaging [7,35].

A summary of the datasets used in this study is provided in Table 2.

### 3.2. Image Preprocessing Techniques

Normalization is a critical component in stabilizing the training dynamics and improving the generalization of deep learning models for chest X-ray (CXR) analysis. Variations in acquisition devices, patient anatomy, and exposure settings introduce substantial heterogeneity across datasets, making preprocessing essential for cross-domain robustness [52]. In this study, three normalization techniques were evaluated systematically: scaling normalization, Z-score normalization, and the proposed adaptive normalization. Representative examples are shown in Figure 1, Figure 2 and Figure 3.

#### 3.2.1. Scaling Normalization

Scaling normalization linearly remaps pixel intensities to a fixed range, typically 0–1 after resizing. A representative example is shown in Figure 1. This method is computationally lightweight and commonly used when imaging conditions are relatively homogeneous [1,53].

However, scaling has notable limitations in multi-institutional CXR settings:It does not correct local contrast variations and is sensitive to outliers caused by acquisition artifacts or metallic implants.It fails to harmonize intensity distributions across scanners, leading to degraded cross-domain generalization [16,39].

Accordingly, scaling normalization was included in this study solely as a baseline preprocessing strategy.

Pixel intensities were linearly rescaled from the original 0–255 range to 0–1 following spatial resizing.

#### 3.2.2. Z-Score Normalization

Z-score normalization standardizes image intensities to zero mean and unit variance (Figure 2), reducing brightness and contrast variations caused by heterogeneous imaging devices and patient populations [29,54]. As a result, it is well-suited for multi-center CXR datasets.

A known limitation is its reliance on approximately Gaussian intensity distributions, which are not always present in clinical CXRs [7]. Nevertheless, Z-score normalization consistently outperforms simple scaling in cross-domain evaluations.

#### 3.2.3. Adaptive Normalization (Proposed Method)

The proposed adaptive normalization method addresses both spatial and intensity variability by combining CDF-guided ROI cropping with histogram standardization. This design is motivated by evidence that localized intensity correction improves diagnostic feature visibility and reduces device-induced domain shifts [16,39,52].

A complete visualization of the pipeline is provided in Figure 3.

Step 1: ROI Localization via CDF-Guided Cropping

To reduce background structures that do not contribute to diagnosis, the cumulative distribution function (CDF) of grayscale-sum values is computed:Horizontal (*x*-axis)—The ROI is extracted between the 5th and 95th percentiles, removing low-density lateral regions that predominantly contain background. This range is selected based on empirical consistency across adult and pediatric CXRs and aligns with findings that lateral regions contribute minimal diagnostic information.Vertical (*y*-axis)—The ROI is retained between the 15th and 95th percentiles, which excludes anatomical noise above the clavicle and reduces variability caused by neck and shoulder structures.

In exploratory experiments across ChestX-ray14, CheXpert, MIMIC-CXR, and Chest-Xray-Pneumonia, these percentile ranges consistently preserved lung apices and costophrenic angles while excluding most neck and shoulder structures. Percentile-based cropping is also more robust to variations in resolution and aspect ratio than fixed-pixel cropping, which facilitates cross-dataset generalization without manual retuning of crop coordinates.

These percentile thresholds were refined through exploratory analysis and are consistent with established approaches for lung-focused cropping in prior studies [39].

Step 2: Histogram-Based Intensity Standardization

To harmonize contrast across datasets with differing exposure characteristics, image histograms were standardized using target statistics derived from normal ChestX-ray14 images:Target mean: μ_target_ = 0.4776 × 255 ≈ 121.8;Target standard deviation: σ_target_ = 0.2238 × 255 ≈ 57.1.

Standardization was applied as defined in Equation (1).(1)$$I_{norm}\left(x,y\right)=\frac{I\left(x,y\right)-\mu_{orig}}{\sigma_{orig}}\cdot \sigma_{target}+\mu_{target}$$
where

$I\left(x,y\right)$: The intensity value of the original image at pixel $\left(x,y\right)$;

$\mu_{orig}$: Mean intensity of the original image;

$\sigma_{orig}$: Standard deviation of the original image;

$\sigma_{target}$: Target standard deviation;

$\mu_{target}$: Target mean;

$I_{norm}\left(x,y\right)$: Normalized pixel value at position $\left(x,y\right)$.

This ensures consistent luminance and contrast across heterogeneous datasets, addressing a key source of domain shift [16].

#### 3.2.4. Summary

Together, these preprocessing techniques ensure that model comparisons are not biased by input variability. While scaling and Z-score normalization provide useful baselines, the proposed adaptive method uniquely addresses both anatomical and intensity-level heterogeneity. By incorporating biologically grounded ROI localization and distribution-aware intensity standardization, the preprocessing pipeline enhances fairness, reproducibility, and cross-domain robustness in clinical deep-learning workflows [52].

### 3.3. Deep Learning Model Architecture

Three convolutional neural network (CNN) architectures were selected to evaluate how different normalization strategies influence performance and generalization: a custom lightweight CNN, EfficientNet-B0, and MobileNetV2. These models represent a spectrum of computational complexity and capacity, which enables a fair and systematic assessment of preprocessing effects across low, medium, and high expressive architectures. Their selection aligns with widely used CNN-based diagnostic pipelines in chest radiography research [20,42,52].

#### 3.3.1. Custom Lightweight CNN

The custom CNN serves as a controlled baseline for isolating the effect of normalization. Its architecture consists of three convolutional layers with ReLU activation, followed by max-pooling, two fully connected layers, and a Softmax output layer for binary classification. With approximately 1.2 million trainable parameters, the model is intentionally lightweight, enabling rapid training and interpretability. This type of architecture has been recommended for applications in resource-limited settings and embedded diagnostic workflows [21,34]. Figure 4 illustrates the structure of this CNN model.

#### 3.3.2. EfficientNet-B0

EfficientNet-B0 employs compound scaling and squeeze-and-excitation blocks, yielding strong performance with a relatively small parameter footprint. Its robustness in thoracic disease classification and cross-domain generalization has been demonstrated in large-scale radiology studies, making it suitable for evaluating normalization under realistic multi-institutional variation [13,37].

#### 3.3.3. MobileNetV2

MobileNetV2 uses inverted residual blocks with depthwise separable convolutions, optimizing it for low-latency inference on embedded and mobile devices. Despite being lightweight, its performance is highly sensitive to preprocessing quality, particularly when trained on heterogeneous public datasets [5,55,56]. Including MobileNetV2 therefore allows the study to evaluate how normalization affects compact architectures deployed in real-time clinical environments.

#### 3.3.4. Model Training Framework

All models were implemented in PyTorch (version 3.13.5) and trained under identical optimization settings to ensure that performance differences arise solely from the normalization methods. Batch-based stochastic training, cross-entropy loss, and unified tracking of accuracy and F1-score were applied across all architectures. The detailed training workflow is summarized using pseudocode in Section 3.4, replacing the previous code listing in accordance with the Reviewer’s recommendations.

### 3.4. Experimental Design

The experimental design was structured to evaluate how different normalization strategies affect chest X-ray classification performance under controlled and comparable conditions. An overview of the experimental pipeline is illustrated in Figure 5.

Four publicly available datasets were considered: ChestX-ray14, CheXpert, MIMIC-CXR, and Chest-Xray-Pneumonia. These datasets differ in patient demographics, acquisition protocols, and imaging devices, making them suitable for cross-domain evaluation [17,18,24].

Each dataset was preprocessed using one of three normalization strategies: scaling normalization, Z-score normalization, and adaptive normalization. Three convolutional neural network architectures were evaluated: a custom lightweight CNN, EfficientNet-B0, and MobileNetV2. This resulted in 36 experimental configurations (4 datasets × 3 normalization methods × 3 models).

All experiments were conducted under identical optimization settings to ensure a fair comparison. Patient-level stratified sampling was applied using an 80% training and 20% validation split. To improve robustness, each configuration was evaluated over three independent runs corresponding to different fixed random seeds (42, 123, and 456), and the reported results represent the mean performance across these runs [57,58]. The detailed training procedure is described using pseudocode in Section 3.4.3.

#### 3.4.1. Training Hyperparameters

All models were trained using identical optimization settings. The hyperparameters used in all experiments are summarized in Table 3.

The training workflow is summarized using pseudocode in Section 3.4.3, replacing the previous code listing in accordance with the Reviewer’s recommendations.

#### 3.4.2. Data Augmentation

To avoid confounding effects on intensity normalization, only mild geometric data augmentation was applied uniformly across all experiments. The augmentation strategy included horizontal flipping, small rotations (±7°), isotropic scaling (0.9–1.1), and translation shifts of up to 5% of the image dimensions.

No brightness, contrast, gamma, or intensity jitter was applied to ensure that intensity statistics remained unaffected by augmentation. This strategy follows established recommendations in medical imaging, where geometric transformations improve robustness without altering radiographic intensity patterns [28,46,47].

#### 3.4.3. Training Workflow Pseudocode

The unified training workflow applied to all model and preprocessing combinations is summarized in Algorithm 1. This pseudocode provides a conceptual description of the optimization process and replaces the previous implementation-level code listing, in accordance with the Reviewer’s recommendations.
**Algorithm 1.** Training Workflow for CXR ClassificationInput
Preprocessed training images
Preprocessed validation images
Neural network model M
Hyperparameters from Table 3Output
Trained model parametersProcedure
Initialize model M with random weights
For each epoch in the allowed maximum number of epochs

Set model M to training mode

For each batch in the training dataset

  Load batch images and labels

  Perform forward pass to obtain predictions

  Compute cross entropy loss

  Compute gradients through backpropagation

  Update model parameters using the Adam optimizer

End batch loop

Set model M to evaluation mode

Compute accuracy and F1 score on the validation dataset
End epoch loopReturn
The final trained model M

This pseudocode ensures that all models are optimized under identical conditions so that performance differences arise exclusively from the normalization methods.

### 3.5. Evaluation Metrics and Performance Formulas

#### 3.5.1. Accuracy

Accuracy measures the proportion of correctly classified samples among all predictions. Although widely used, accuracy may become unreliable when dealing with imbalanced medical datasets, which is frequently observed in chest X-ray classification [8]. For this reason, accuracy is reported together with more stable metrics to ensure a balanced evaluation. The formula is provided in Equation (2).(2)$$\mathrm{Accuracy}=\frac{TP+TN}{TP+FP+TN+FN}$$

#### 3.5.2. F1-Score

The F1-score is the harmonic mean of precision and recall. It provides a balanced perspective by considering both false positives and false negatives. F1-score is particularly valuable in medical imaging because many datasets contain imbalanced class distributions with fewer positive cases [8]. Its definition appears in Equation (3), with the expressions for precision and recall shown in Equation (4).(3)$$F1=2\cdot \frac{\mathrm{Precision}\cdot \mathrm{Recall}}{\mathrm{Precision}+\mathrm{Recall}}$$
where(4)$$\mathrm{Precision}=\frac{TP}{TP+FP},\;\;\;\;\;\;\;\;\;\;\;\;\;\;\;\;\mathrm{Recall}=\frac{TP}{TP+FN}$$

In this study, the F1-score is a central evaluation metric, especially for imbalanced datasets such as CheXpert and MIMIC-CXR. To improve reliability, In this study, we report the mean and standard deviation of F1-scores obtained from multiple independent runs using different fixed random seeds.

#### 3.5.3. Sensitivity and Specificity

Sensitivity reflects the proportion of abnormal cases correctly detected by the model. Specificity reflects the proportion of normal cases correctly identified. These measures complement accuracy and F1-score by providing a more detailed analysis of performance under class imbalance, which is crucial for clinical reliability [8]. The formulas are given in Equation (5).(5)$$\mathrm{Sensitivity}=\frac{TP}{TP+FN},\;\;\;\;\;\;\;\mathrm{Specificity}=\frac{TN}{TN+FP}$$

Sensitivity helps evaluate the model’s ability to detect disease, while specificity measures its capability to avoid false alarms. Together, they provide additional insight into the diagnostic behavior of the models across datasets with varying prevalence rates.

### 3.6. Statistical Significance Testing

To assess whether the performance differences observed across normalization methods and model architectures were statistically meaningful, a three-stage non-parametric testing procedure was applied. This approach follows established recommendations for model comparison in medical image analysis, where performance metrics typically violate assumptions of normality [8,42].

First, the Friedman test was used to examine overall differences across all normalization–model combinations evaluated on multiple datasets. The test is appropriate for repeated-measures settings in which identical classifiers are compared across several conditions without assuming Gaussian distributions [8]. The Friedman test yielded significant results with *p*-values below 0.05, indicating that at least one method differed from the others.

Following this outcome, the Nemenyi post hoc test was applied to identify which pairs of normalization approaches exhibited statistically significant differences. The Nemenyi test is recommended for pairwise comparisons after Friedman analysis, particularly for experiments involving multiple algorithms evaluated under identical settings [22].

Finally, the Wilcoxon signed-rank test was employed to provide fine-grained paired comparisons of F1-scores between preprocessing methods under identical model architectures and data splits. This test is widely recommended for paired evaluations in cross-validation pipelines and is robust to the non-normal distribution of performance scores [8,59].

The results of the Wilcoxon test are summarized in Table 4.

This statistical analysis further supports the robustness of the observed performance differences, suggesting that the improvements associated with adaptive normalization are unlikely to be attributable to random variation.

## 4. Experimental Results

This section reports a comprehensive evaluation of the three normalization strategies, scaling, Z-score, and adaptive normalization, across four chest X-ray datasets and three convolutional architectures. Composite figures (Figure 6, Figure 7 and Figure 8) and reorganized performance tables (Table 5, Table 6 and Table 7) were used to summarize the validation accuracy, loss, F1-score, and interaction effects between normalization and model design.

### 4.1. Accuracy Analysis

Table 5 and Figure 6 summarize the validation accuracy across datasets and model architectures. Across three of the four datasets, adaptive normalization generally maintained or improved accuracy relative to scaling and Z-score normalization, particularly on ChestX-ray14, CheXpert, and Chest-Xray-Pneumonia.

MIMIC-CXR constitutes the main exception, where adaptive normalization yielded slightly lower accuracy than Z-score normalization, reflecting the dataset’s substantial acquisition and intensity heterogeneity.

Overall, the accuracy results indicate that adaptive normalization improves cross-dataset generalization when contrast characteristics are relatively stable, while its effectiveness remains dataset-dependent.

Each block corresponds to one dataset (ChestX-ray14, CheXpert, MIMIC-CXR, Chest X-ray Pneumonia). Bars show the mean validation accuracy over three independent runs for each model (CNN, EfficientNet-B0, MobileNetV2) and normalization strategy (Scaling, Z-score, Adaptive). “Adaptive” refers to the pipeline of CDF-based ROI cropping followed by histogram standardization using the fixed target mean and standard deviation; “Scaling” corresponds to min–max intensity scaling; and “Z-score” uses global dataset-level statistics.

### 4.2. Loss Analysis

Validation loss results (Table 6 and Figure 7) largely mirror the accuracy trends. In most configurations, adaptive normalization achieved lower or comparable validation loss compared with scaling and Z-score normalization, indicating more stable optimization behavior.

For MIMIC-CXR, adaptive normalization did not provide a consistent loss advantage, again highlighting the challenges posed by strong inter-institution variability.

These findings suggest that adaptive normalization generally stabilizes training dynamics, although its benefits are constrained in datasets with highly heterogeneous acquisition conditions.

Each block corresponds to one dataset (ChestX-ray14, CheXpert, MIMIC-CXR, Chest X-ray Pneumonia). Bars represent the mean validation loss over three runs for each combination of model (CNN, EfficientNet-B0, MobileNetV2) and normalization strategy (Scaling, Z-score, Adaptive). Lower loss values indicate better calibration stability and convergence.

### 4.3. F1-Score Analysis

Table 7 and Figure 8 report F1-scores across all configurations, providing a balanced evaluation under class imbalance. Adaptive normalization achieved the highest F1-scores on datasets with more consistent intensity characteristics, with the strongest gains observed on Chest-Xray-Pneumonia.

In contrast, Z-score normalization remained competitive or superior on MIMIC-CXR, where broader distributional shifts reduced the effectiveness of histogram-based standardization.

Statistical testing using the Wilcoxon signed-rank test confirmed that adaptive normalization significantly outperformed scaling and Z-score normalization in most dataset–model combinations (*p* < 0.01), except for MIMIC-CXR.

Each block corresponds to one dataset (ChestX-ray14, CheXpert, MIMIC-CXR, Chest X-ray Pneumonia). Bars represent the mean validation F1-score over three runs for each combination of model (CNN, EfficientNet-B0, MobileNetV2) and normalization strategy (Scaling, Z-score, Adaptive). Higher F1-scores indicate better classification performance and calibration.

Each row shows one test image that was misclassified under Z-score normalization but correctly classified after applying the proposed adaptive normalization. The left column presents the original CXR after Z-score normalization, where low contrast and background dominance contribute to incorrect predictions. The right column shows the same images after CDF-guided cropping and histogram standardization, which enhanced the visibility of lung fields and reduced non-diagnostic background variation. Ground-truth labels and model predictions from MobileNetV2 are indicated in each panel.

Representative examples of misclassified Chest X-ray images before and after normalization are shown in Figure 9.

### 4.4. Ablation Study: Effect of Cropping and Histogram Standardization

To isolate the contribution of each component in the proposed adaptive normalization pipeline, we conducted an ablation study comparing four conditions: (A) Z-score, (B) cropping only, (C) histogram standardization only, and (D) the full adaptive pipeline. Results on the Chest-Xray-Pneumonia dataset using MobileNetV2 are summarized in Table 8. In this ablation, Z-score normalization serves as the baseline condition, representing input images without either adaptive component (cropping or histogram standardization). For clarity, conditions (A)–(D) correspond directly to the four rows in Table 8.

Histogram standardization produced the largest standalone improvement, increasing the F1-score from 0.85 to 0.88 by harmonizing contrast and suppressing global intensity variability. Cropping yielded a smaller but consistent benefit (0.86) by increasing the lung-to-image ratio and removing non-diagnostic background regions. The full adaptive pipeline achieved the highest performance (0.89), confirming that cropping and histogram standardization are complementary rather than redundant. These results indicate that both components contribute meaningfully to the overall gain, with histogram standardization exerting a stronger individual effect for lightweight architectures such as MobileNetV2.

### 4.5. Interaction Between Architecture and Normalization

MobileNetV2 exhibited the strongest synergy with adaptive normalization, consistently outperforming or matching the other architectures on datasets with clearer and more uniform intensity characteristics such as ChestX-ray14, CheXpert, and Chest-Xray-Pneumonia. This complementarity arises from several architectural features that are particularly responsive to intensity standardization.

First, depthwise-separable convolutions are highly sensitive to fluctuations in local pixel distribution. Adaptive normalization reduces these fluctuations through region-focused cropping and histogram standardization, leading to more consistent feature activation. Second, the linear bottlenecks in MobileNetV2 benefit from reduced background variation, enabling the model to emphasize diagnostically meaningful structures within the lung fields. The adaptive cropping step further reinforces this effect by focusing the input on anatomically relevant regions.

In datasets where contrast characteristics are relatively stable, these factors collectively support smoother optimization, more stable validation curves, and improved detection of lung opacity patterns. This behavior is also clinically meaningful, as consistent contrast normalization improves the visibility of subtle radiographic abnormalities.

In contrast, for datasets with substantial acquisition variability such as MIMIC-CXR, the advantages of adaptive normalization are less apparent, reflecting the challenges posed by broad cross-institution intensity shifts. Overall, these findings highlight the importance of aligning normalization strategies with model architectures that are particularly receptive to intensity standardization, especially in medical imaging tasks affected by domain variability.

## 5. Discussion

The experimental results demonstrate that adaptive normalization improves or maintains validation performance relative to scaling and Z-score normalization on three of the four benchmark datasets. This finding is consistent with prior evidence highlighting the importance of intensity normalization and local contrast enhancement in medical image analysis, particularly in multi-center or multi-device settings [7,29,35,39]. By combining percentile-based ROI cropping with histogram standardization, the pipeline reduces non-diagnostic background variation while preserving diagnostically relevant lung structures, leading to more consistent model behavior under moderate domain shifts.

The contrast between the substantial gains observed on ChestX-ray14, CheXpert, and Chest-Xray-Pneumonia and the limited improvements on MIMIC-CXR highlights an important limitation. MIMIC-CXR exhibited pronounced variability in acquisition settings and device characteristics [17,18], weakening the assumptions required for effective histogram-based intensity standardization. Similar observations have been reported in multi-site MRI and radiomics studies, where no single normalization technique consistently dominates under extreme inter-site heterogeneity [7,29,35]. These findings indicate that normalization strategies should be selected according to the expected degree of domain variability rather than applied uniformly across all deployment scenarios. In addition, because pediatric thoraces are proportionally smaller, fixed percentile thresholds may remove relatively larger apical regions, representing a potential source of anatomical bias when adult-derived cropping parameters are applied to pediatric CXRs. This study did not include a dedicated pediatric subgroup analysis; therefore, future work should explicitly validate and, if necessary, re-tune cropping parameters for pediatric populations.

From an architectural perspective, MobileNetV2 combined with adaptive normalization yielded the most reliable performance on datasets with relatively uniform intensity characteristics. This behavior is consistent with the known sensitivity of depthwise-separable convolutions and linear bottlenecks to input distribution consistency [55]. In lightweight architectures, suppressing irrelevant background variation allows representational capacity to focus on subtle parenchymal abnormalities, which is critical for detecting thoracic diseases such as pneumonia and COVID-19 in resource-limited settings [19,20,34]. These results support the view that preprocessing pipelines and model architectures should be co-designed to achieve optimal generalization.

The evaluation framework employed in this study follows established recommendations for rigorous assessment in medical imaging. Reporting F1-scores alongside complementary metrics provides a more reliable evaluation under class imbalance [60]. Statistical significance was assessed using non-parametric tests, including the Friedman–Nemenyi and Wilcoxon signed-rank tests, which are widely endorsed for classifier comparison across multiple datasets [61]. The resulting *p*-values indicate that the improvements achieved by adaptive normalization were both numerically and statistically meaningful in most scenarios.

Two primary clinical implications emerge from these findings. First, improved robustness under moderate domain shift supports the deployment of CXR-based decision-support systems across institutions with heterogeneous acquisition pipelines, complementing recent advances in efficient and multi-label CXR modeling [6,22,26,45]. Second, the computational efficiency of MobileNetV2, even with the additional preprocessing step, enables near-real-time inference suitable for triage or screening workflows in constrained healthcare environments [34,41].

Adaptive normalization introduced a modest computational overhead. Empirically, the pipeline increased training time by approximately 5.2 ms per batch (+6.3%), with 3.5 ms attributed to CDF-based ROI cropping and 1.7 ms to histogram standardization. This overhead is negligible for offline training and does not affect inference latency because normalization is applied once during preprocessing. Although acceptable in the present setting, such overhead may become relevant for large-scale pretraining or continuous-learning pipelines. Moreover, this study evaluated only four publicly available datasets; performance on unseen clinical cohorts from new scanners, populations, or institutions remains to be systematically assessed. No clinician-in-the-loop assessment or external prospective validation was conducted; therefore, conclusions are limited to retrospective evaluations on publicly available benchmarks.

These limitations motivate several future research directions. Integrating adaptive normalization into federated or source-free domain adaptation frameworks may further improve cross-site robustness while preserving data privacy [14,15]. In addition, combining the pipeline with self-supervised or contrastive pretraining on large-scale unlabeled radiograph repositories could enhance sample efficiency and downstream performance [21,57,58]. Finally, explainability techniques such as Grad-CAM may help elucidate how normalization reshapes salient features, supporting interpretability and clinical adoption [13,56].

Overall, our cross-dataset evaluation reinforces that preprocessing is a critical component of clinically deployable deep learning pipelines. When aligned with dataset characteristics and model architecture, adaptive normalization provides a practical approach for improving generalization in chest X-ray classification while maintaining computational efficiency.

## 6. Conclusions

This study examined how three preprocessing strategies, scaling, Z-score normalization, and a proposed adaptive normalization pipeline, affect the generalization performance of deep learning models for chest X-ray classification across four benchmark datasets and three convolutional architectures. The findings show that adaptive normalization consistently improves or maintains validation accuracy, loss, and F1-score relative to conventional methods on three of the four datasets, with the strongest performance gains observed when combined with the MobileNetV2 architecture. These improvements reflect more consistent localization of the lung fields and better standardization of intensity distributions, which collectively contribute to more stable optimization and reduced overfitting.

The limited improvements observed on MIMIC-CXR highlight that the effectiveness of normalization depends strongly on the underlying acquisition heterogeneity. In datasets characterized by substantial cross-institution variation, global histogram-based adjustments alone are insufficient, underscoring the need for normalization strategies that explicitly account for broad domain shifts.

The contributions of this study are threefold. First, it provides a cross-dataset, cross-architecture benchmarking framework for systematically evaluating normalization strategies in chest X-ray classification. Second, it offers a statistically grounded comparison using non-parametric tests across multiple metrics, demonstrating that the performance differences are both numerically and statistically meaningful. Third, it demonstrates that an efficient adaptive normalization pipeline integrates well with lightweight architectures such as MobileNetV2, making it suitable for deployment in resource-constrained clinical environments.

Future work will extend this pipeline to more diverse real-world cohorts and explore its integration with federated and source-free domain adaptation frameworks, as well as self-supervised pretraining on large unlabeled CXR collections. Another important direction is to combine adaptive normalization with explainable AI techniques to visualize how preprocessing influences salient regions, thereby facilitating clinical validation and trustworthy deployment of deep learning models in radiographic diagnostics.

## Figures and Tables

**Figure 1 jimaging-12-00014-f001:**
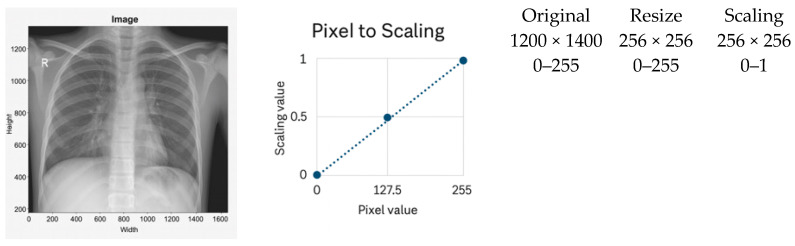
Example of scaling normalization applied after image resizing. The dashed line represents the linear mapping from pixel intensity values (0–255) to the normalized range (0–1), and the dots indicate representative pixel values before and after scaling.

**Figure 2 jimaging-12-00014-f002:**
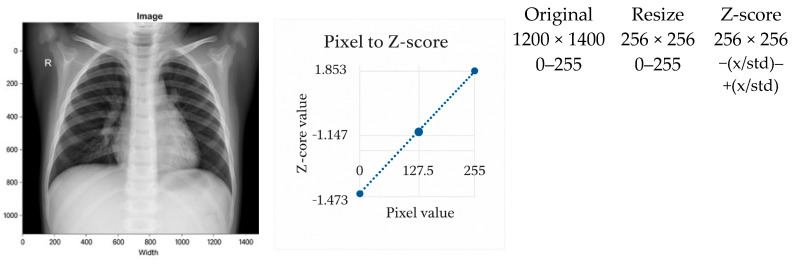
Example of Z-score normalization applied after image resizing. The dashed line represents the linear transformation from pixel intensity values (0–255) to standardized Z-score values, while the dots indicate representative pixel values before and after normalization.

**Figure 3 jimaging-12-00014-f003:**
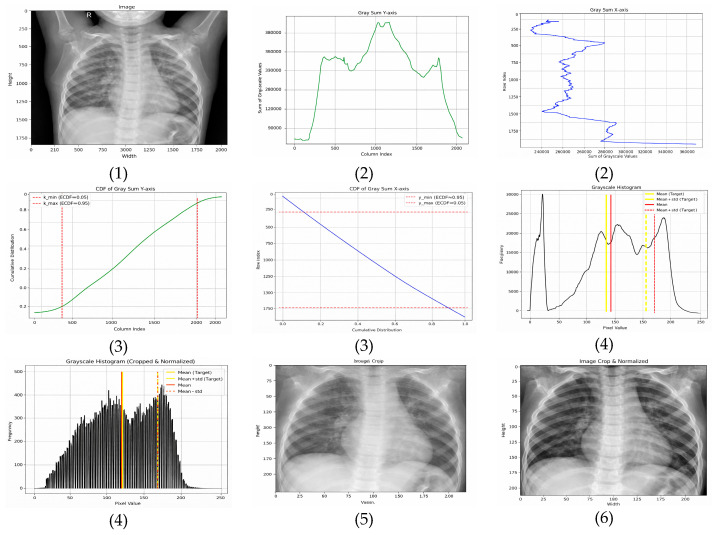
Visualization of the adaptive normalization pipeline. (**1**) Original CXR image, (**2**) grayscale-sum profiles computed along the horizontal (column-wise) and vertical (row-wise) axes, (**3**) CDF-based percentile thresholds, where the solid colored lines denote the selected lower and upper cutoffs used for ROI cropping, (**4**) grayscale histograms before and after normalization, (**5**) cropped ROI image, and (**6**) final normalized image.

**Figure 4 jimaging-12-00014-f004:**
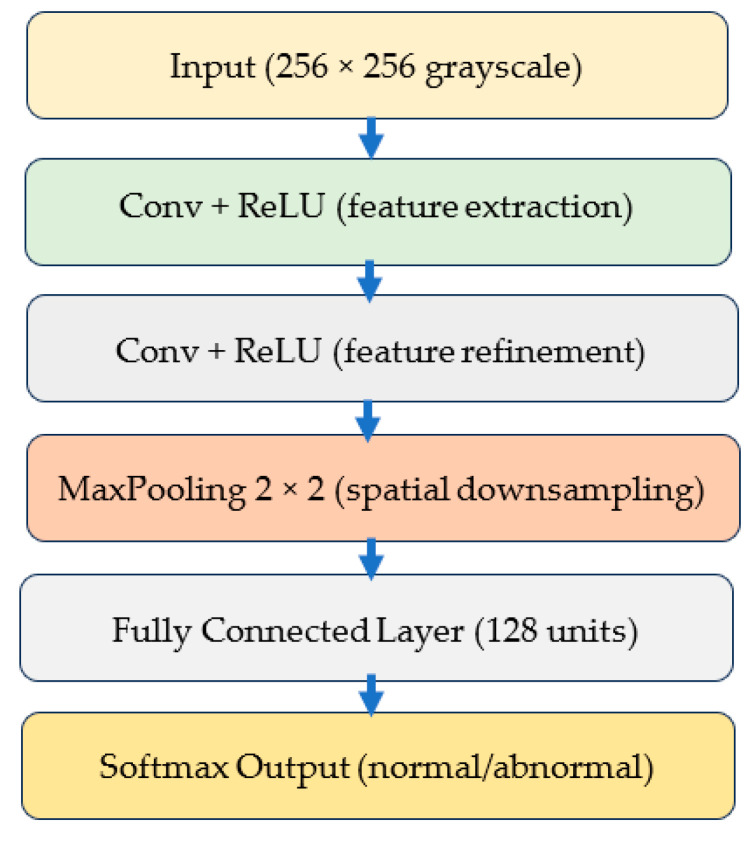
Conceptual representation of the lightweight CNN used in this study. The arrows indicate the sequential data flow through the network layers, while the different colors denote distinct functional stages, including feature extraction, feature refinement, spatial downsampling, and classification.

**Figure 5 jimaging-12-00014-f005:**
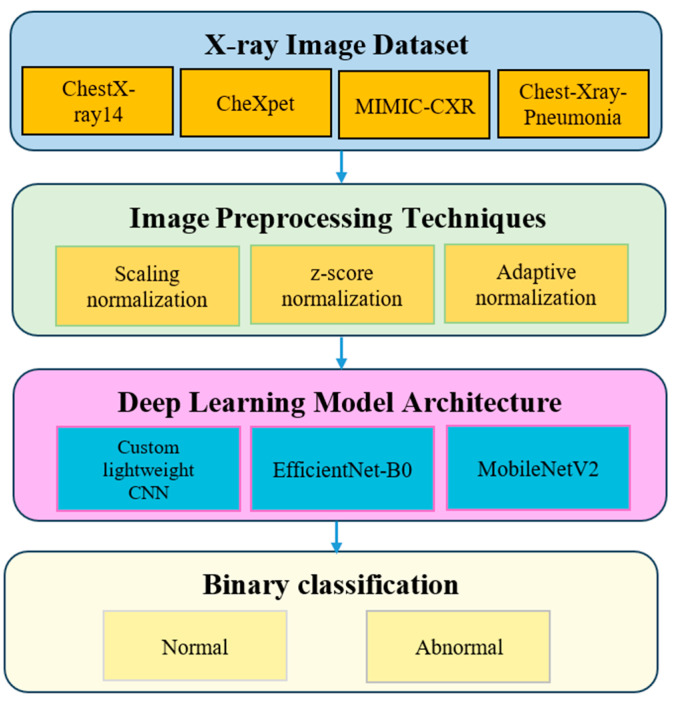
Overview of the experimental pipeline illustrating datasets, normalization strategies, model architectures, and binary classification task. The arrows indicate the sequential flow of data through the experimental pipeline, while the different colors are used to distinguish major stages, including dataset sources, image preprocessing techniques, deep learning model architectures, and the final classification task.

**Figure 6 jimaging-12-00014-f006:**
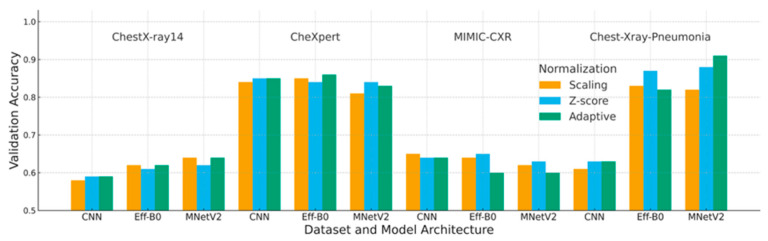
Composite validation accuracy across datasets and model architectures.

**Figure 7 jimaging-12-00014-f007:**
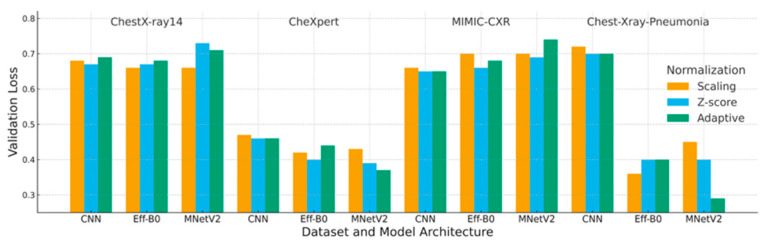
Composite validation loss across datasets and model architectures.

**Figure 8 jimaging-12-00014-f008:**
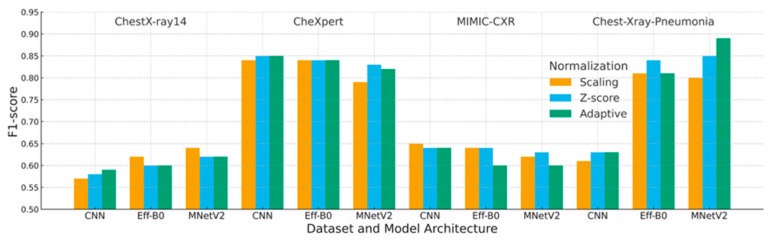
Composite validation F1-score across datasets and model architectures.

**Figure 9 jimaging-12-00014-f009:**
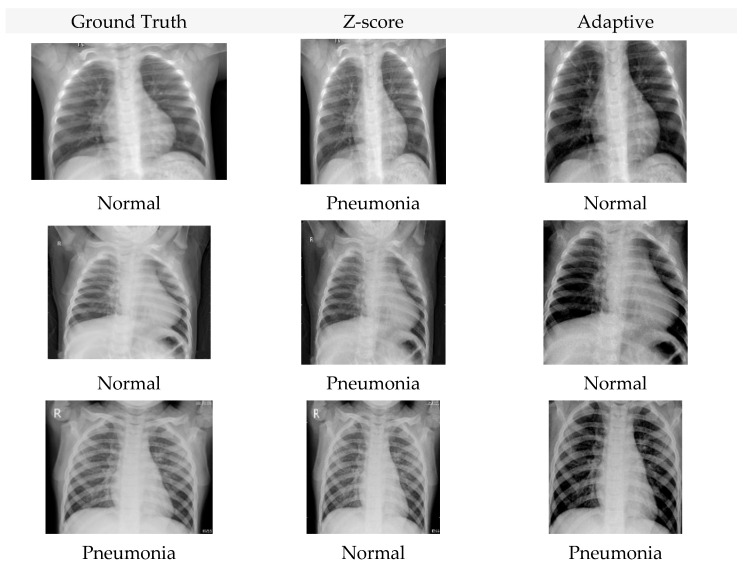
Misclassified Chest-Xray-Pneumonia cases before and after adaptive normalization. Examples show cases incorrectly classified under Z-score normalization but correctly classified after applying the proposed adaptive normalization.

**Table 1 jimaging-12-00014-t001:** Comparative Summary of Deep Learning, Normalization, and Augmentation Studies in Chest X-ray/CT Imaging (2020–2025). *Reorganized into three groups as requested by the Reviewers: (A) Deep Learning Architectures for CXR/CT Classification, (B) Normalization and Preprocessing Approaches, and (C) Data Augmentation and Transferability Studies*.

A. Deep Learning Architectures for CXR/CT Classification.										
**Authors**	**Model/Architecture**	**Technique/Approach**			**Dataset**			**Metrics**		**Key Highlights**
[13]	Inf-Net	Semi-supervised infection segmentation with reverse and edge attention			COVID-SemiSeg			Dice, Sens., Spec.		First semi-supervised CT infection segmentation dataset and model
[1]	ResNet-18 + CAM	Patch-based semi-supervised CXR learning			COVIDx, RSNA			Accuracy, AUC		Strong performance with limited labeled CXRs
[19]	VGG19, DenseNet, Inception	Transfer learning with augmentation			Chest X-ray Pneumonia			Accuracy, F1		Demonstrated impact of augmentation and TL
[40]	COVIDNet-CT	COVID-specific CNN			COVIDx-CT			Accuracy, Sens.		High-performing CT-based classifier
[42]	DarkCOVIDNet	CNN for binary/multi-class COVID-19 detection			COVID-19 X-ray			Accuracy, F1		Early high-performing CXR classifier
[33]	MobileNetV2 + SqueezeNet	Fuzzy preprocessing with metaheuristic optimization			COVID (Cohen)			Accuracy, F1		Fusion of deep and fuzzy features
[20]	COVID-Net	Machine-designed CNN			COVIDx			Accuracy, Sens.		Transparent, explainable CNN design
[31]	InceptionResNetV2 + BiLSTM	Hybrid deep features (GLCM/LBP + CNN)			COVIDx			Accuracy, AUC		Outperformed CNN-only baselines
[5]	LT-ViT	Label-token vision transformer			CheXpert, CXR14			AUC		Interpretable ViT via explicit label tokens
[11]	ResNet-based	Style and histogram normalization pipeline			Multi-hospital CXR			AUC, Accuracy		Cross-device robust AI pipeline
[10]	EfficientNet-B4	CLAHE + augmentation			CXR14, PadChest			F1, AUC		High robustness with strong AUC
[34]	Lightweight CNN	Embedded-optimized CNN			CXR14			Accuracy, Sens.		Real-time classification on heterogeneous devices
[43]	Lightweight CNN	Edge-oriented deployment			CXR14			Accuracy, Precision, Sens.		Ultra-light model with near-ResNet performance
[44]	Reconstruction-based CNN + XGBoost	GAN reconstruction, segmentation + radiomics			CXR14			AUC, Accuracy		Pathology-aware reconstruction and classification
[45]	EHTNet (Hybrid CNN–Transformer)	Explainable hybrid transformer for lung diseases			CXR14			Accuracy, AUC		Hybrid CNN–ViT architecture with attention-based explanations
[2]	ViT + Metaheuristics	ViT optimized with PSO/GWO for severity and multimodal fusion			CXR + CT			Accuracy, Sens.		ViT tuned via metaheuristic optimizers
This Study	CNN, EfficientNet-B0, MobileNetV2	Adaptive normalization (CDF cropping + histogram standardization)			CXR14, CheXpert, MIMIC-CXR, Pediatric			Accuracy, F1		A systematic cross-dataset, cross-architecture benchmarking of normalization strategies under controlled sampling.
B. Normalization and Preprocessing Approaches										
**Authors**	**Method**		**Dataset**				**Metrics**		**Key Highlights**	
[39]	Localized energy-based normalization		Chest radiography				Sens., Spec.		Foundational localized contrast standardization for CXRs	
[30]	Adaptive Instance Normalization (AdaIN)		Natural + style datasets				Style metrics		Basis for modern adaptive, style-aware normalization	
[14]	Self-attentive spatial adaptive normalization		Radiography/CT				Dice, IoU		Spatially adaptive normalization for cross-modality DA	
[11]	Histogram + style normalization pipeline		Multi-hospital CXR				Accuracy, AUC		Reduced device-driven variation across machines	
[16]	Exposure-region adaptive histogram equalization		CXR				PSNR, entropy		Modern exposure-region contrast enhancement method	
This Study	CDF-guided cropping + histogram standardization		CXR14, CheXpert, MIMIC-CXR				Accuracy, F1		Joint ROI and intensity standardization for multi-source CXRs	
C. Data Augmentation and Transferability Studies										
**Authors**	**Focus**			**Dataset(s)**		**Metrics**			**Key Highlights**	
[28]	Systematic review of data augmentation in medical imaging			Multi-modal medical		Narrative			Identified geometric transforms as backbone of medical DL augmentation	
[46]	Augmentation for clinical deterioration prediction			CXR		Accuracy, AUC			Showed rotation and flipping improve robustness in clinical prediction	
[47]	Augmentation for TB/COVID robustness			Multiple CXR datasets		AUC, Precision			Brightness and gamma augmentations improve cross-dataset transferability	
[33]	Fuzzy preprocessing + augmentation			CXR (COVID)		Accuracy, F1			Demonstrated that fuzzy preprocessing with augmentation enhances performance	

**Table 2 jimaging-12-00014-t002:** Summary of Chest X-ray Datasets Used in Normalization Evaluation.

Dataset	Image Count	Images Used	Patients	Original Labels/Classes	Classes Used
ChestX-ray14	112,120	16,000	30,805	14	2
CheXpert	224,316	16,000	65,240	14	2
MIMIC-CXR	377,110	16,000	227,827	14	2
Chest-Xray-Pneumonia	5863	5863	Pediatric only	3	2

**Table 3 jimaging-12-00014-t003:** Hyperparameters used for all experiments.

Hyperparameter	Value
Optimizer	Adam
Learning rate	1 × 10^−4^
Batch size	100
Maximum epochs	20
Weight decay	1 × 10^−5^
Train validation split	80%/20%
Random seeds	42, 123, 456
Loss function	Cross entropy

**Table 4 jimaging-12-00014-t004:** Wilcoxon Signed-Rank Test Results (n = 12 paired dataset × model configurations).

Comparison	*p*-Value	Significance
Adaptive vs. Z-score	0.0078	Significant (*p* < 0.01)
Adaptive vs. Scaling	0.0039	Significant (*p* < 0.01)
Z-score vs. Scaling	0.0781	Not significant (*p* > 0.05)

**Table 5 jimaging-12-00014-t005:** Summary of Validation Accuracy Across Datasets and Configurations.

Dataset	Deep Learning	Scaling	Z-Score	Adaptive
ChestX-ray14	CNN	0.58	0.59	0.59
	EfficientNet-B0	0.62	0.61	0.62
	MobileNetV2	0.64	0.62	0.64
CheXpert	CNN	0.84	0.85	0.85
	EfficientNet-B0	0.85	0.84	0.86
	MobileNetV2	0.81	0.84	0.83
MIMIC-CXR	CNN	0.65	0.64	0.64
	EfficientNet-B0	0.65	0.65	0.60
	MobileNetV2	0.62	0.63	0.60
Chest-Xray-Pneumonia	CNN	0.61	0.63	0.63
	EfficientNet-B0	0.83	0.87	0.82
	MobileNetV2	0.82	0.88	0.91

Cells highlighted in green indicate the highest validation accuracy within each dataset and model configuration. “Adaptive” refers to the proposed adaptive normalization pipeline consisting of CDF-based ROI cropping followed by histogram standardization using fixed target mean and standard deviation. “Scaling” corresponds to min–max scaling to the [0, 1] range, while “Z-score” normalization uses the global dataset-level mean and standard deviation. All results are averaged over three runs with fixed random seeds and identical stratified data splits.

**Table 6 jimaging-12-00014-t006:** Summary of Validation Loss Across Datasets and Configurations.

Dataset	Preprocessing	Scaling	Z-Score	Adaptive
ChestX-ray14	CNN	0.68	0.67	0.69
	EfficientNet-B0	0.66	0.67	0.68
	MobileNetV2	0.66	0.73	0.71
CheXpert	CNN	0.47	0.46	0.46
	EfficientNet-B0	0.42	0.40	0.44
	MobileNetV2	0.43	0.39	0.37
MIMIC-CXR	CNN	0.66	0.65	0.65
	EfficientNet-B0	0.70	0.66	0.68
	MobileNetV2	0.70	0.69	0.74
Chest-Xray-Pneumonia	CNN	0.72	0.70	0.70
	EfficientNet-B0	0.36	0.40	0.40
	MobileNetV2	0.45	0.40	0.29

Lower loss values indicate better model calibration and convergence. All loss values represent averages over three experimental runs.

**Table 7 jimaging-12-00014-t007:** Summary of Validation F1-scores Across Datasets and Configurations.

Dataset	Preprocessing	Scaling	Z-Score	Adaptive
ChestX-ray14	CNN	0.57	0.58	0.59
	EfficientNet-B0	0.62	0.60	0.60
	MobileNetV2	0.64	0.62	0.62
CheXpert	CNN	0.84	0.85	0.85
	EfficientNet-B0	0.84	0.84	0.84
	MobileNetV2	0.79	0.83	0.82
MIMIC-CXR	CNN	0.65	0.64	0.64
	EfficientNet-B0	0.64	0.64	0.60
	MobileNetV2	0.62	0.63	0.60
Chest-Xray-Pneumonia	CNN	0.61	0.63	0.63
	EfficientNet-B0	0.81	0.84	0.81
	MobileNetV2	0.80	0.85	0.89

Higher F1-scores indicate better model calibration and classification performance. All values represent averages over three experimental runs.

**Table 8 jimaging-12-00014-t008:** Ablation Study Results (MobileNetV2, Chest-Xray-Pneumonia).

Method	Cropping	Histogram	F1-Score
Z-score	✗	✗	0.85
Cropping only	✓	✗	0.86
Histogram only	✗	✓	0.88
Adaptive	✓	✓	0.89

All values represent the mean of three runs using identical stratified splits and the hyperparameters described in Section 3.4. ✓ indicates that the corresponding component is applied, while ✗ indicates that it is not applied.

## Data Availability

The datasets used in this study are publicly available from established benchmark repositories (ChestX-ray14, CheXpert, MIMIC-CXR, and Chest X-ray Pneumonia). Due to licensing terms and data governance policies of these repositories, the processed data used in this study are not redistributed publicly but are available from the corresponding author upon reasonable request.

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
