# Peer review of "Adaptive Normalization Enhances the Generalization of Deep Learning Model in Chest X-Ray Classification"

_2313-433X, 2025, doi:10.3390/jimaging12010014_

Round 1
Reviewer 1 Report (Previous Reviewer 3)
Comments and Suggestions for Authors
Novelty and Contribution: The manuscript provides a systematic empirical comparison of three normalization strategies for CXR classification—scaling, Z-score, and an adaptive percentile-based cropping with histogram standardization. While normalization is a well-studied topic in CXR analysis, your contribution lies in the breadth of the evaluation (four datasets × three CNN models) and the proposal of a combined ROI-cropping + histogram-harmonization workflow. This contribution is incremental rather than conceptually novel, because both histogram standardization and ROI-based cropping are established techniques in medical imaging; however, the integration and systematic benchmarking across multiple datasets does constitute a meaningful applied contribution. The paper should tone down claims of originality and emphasize the value of comprehensive comparative evaluation rather than methodological innovation.
Methodological Clarity: The methodology is described in extensive detail, sometimes excessively so. Sections on MFCC-like intensity discussions, CDF percentile decisions, and CNN basics are much longer than necessary for a J. Imaging audience. However, the adaptive pipeline is explained clearly, with equations and diagrams that help readers understand the steps. What is missing is a justification of percentile thresholds, explanation of why these ranges generalize across datasets, and a discussion of the computational overhead added by batch-wise histogram adjustment. A concise high-level architectural diagram of the entire pipeline (normalization → model → evaluation) would strengthen reader comprehension.
Experimental Design and Evaluation: The experimental setup is thorough and includes four datasets (ChestX-ray14, CheXpert, MIMIC-CXR, Pneumonia) and three CNN architectures. The use of identical training hyperparameters and multiple seeds is appropriate, and the inclusion of accuracy, F1, loss, sensitivity, and statistical tests is a strength. However, the evaluation is limited by two factors: (1) binary classification only, which simplifies the real complexity of CXR diagnosis; and (2) absence of external testing on unseen institutions beyond the sampled subsets. The strongest limitation is that most improvements are small (often 1–2%) and sometimes inconsistent, particularly on MIMIC-CXR. A dedicated ablation of “cropping only” vs “histogram only” would help isolate which component contributes to the gains.
Interpretation of Results: The interpretation correctly notes that adaptive normalization helps most datasets except MIMIC-CXR, but the discussion remains descriptive rather than analytical. For example, the paper does not explore why histogram standardization can fail under extreme acquisition variability, nor how specific pathologies influence performance under different normalization schemes. Additionally, the discussion would benefit from an exploration of potential bias (e.g., pediatric vs adult), and how cropping affects anatomical regions such as costophrenic angles or apical zones. A qualitative visualization of misclassified cases before/after normalization would significantly strengthen interpretability.
Writing Quality and Structure: The manuscript is organized clearly and includes consistent figures, tables, and examples. Nevertheless, the text is too long and repetitive, especially in Sections 2 and 3, which read like a tutorial rather than a research article. Much of the background on CNNs, normalization, and augmentation could be shortened without losing clarity. The conclusions section is strong, but the introduction could better emphasize the specific gaps this work addresses. Figures are clear, though some (e.g., multi-panel composites) could be simplified.
Related Work and Positioning: The related-work section is extensive and detailed but sometimes unfocused. It mixes preprocessing, architectures, and domain adaptation papers without tightly linking them to the exact problem addressed. The manuscript also misses important recent works on self-supervised CXR representations, vision transformers, and foundation models, which are increasingly central to generalization research. Positioning your study more clearly as a normalization benchmarking contribution would resolve the current overstated novelty.
Overall Assessment: This is a solid applied study, with strong empirical depth but limited novelty. The normalization pipeline works well on datasets with moderate variability, and the experiments are carefully run. However, the manuscript requires substantial condensation, clearer positioning relative to prior work, and deeper interpretive analysis of why and when adaptive normalization helps.
Author Response
|
Response to Reviewers’ Comments Manuscript Title: Adaptive Normalization Enhances the Generalization of Deep Learning Model in Chest X-ray Classification
|
||
|
1. Summary |
|
|
|
Thank you very much for taking the time to review this manuscript. Please find the detailed responses below and the corresponding revisions/corrections highlighted/in track changes in the re-submitted files. In response to the comments, we have thoroughly revised the manuscript to: (i) clearly position the study as a systematic benchmarking and applied preprocessing evaluation, rather than a novel algorithmic contribution; (ii) condense and refocus methodological descriptions to avoid tutorial-style exposition; (iii) include a dedicated ablation study isolating the effects of ROI cropping and histogram standardization; (iv) provide explicit justification of percentile thresholds used for ROI cropping; (v) report and discuss the computational overhead introduced by the adaptive pipeline; and (vi) strengthen the interpretation of results, including dataset-specific limitations and potential anatomical bias (e.g., pediatric vs. adult CXRs), supported by qualitative error visualizations.
All revisions have been incorporated into the revised manuscript and are highlighted using track changes.
|
||
|
2. Questions for General Evaluation |
Reviewer’s Evaluation |
Response and Revisions |
|
Does the introduction provide sufficient background and include all relevant references? |
Must be improved |
The Introduction was restructured to emphasize the specific research gap in normalization benchmarking and cross-dataset robustness (Section 1, pp. 1–2).
|
|
Are all the cited references relevant to the research?
|
Must be improved |
References were reorganized and grouped thematically to improve focus and relevance (Section 2.6; Table 1, pp. 6–7).
|
|
Is the research design appropriate?
|
Must be improved |
The experimental design was clarified by emphasizing controlled data splits, identical hyperparameters, and multi-seed evaluation (Section 3.4, pp. 13–15).
|
|
Are the methods adequately described?
|
Must be improved |
Excessively detailed tutorial-style content was condensed, while key methodological justifications were added (Sections 2.4–2.5 and 3.2–3.4, pp. 9–15).
|
|
Are the results clearly presented?
|
Must be improved |
Results were reorganized with clearer composite figures and tables, and qualitative error analysis was added (Section 4, pp. 17–22).
|
|
Are the conclusions supported by the results?
|
Must be improved |
Conclusions were explicitly aligned with empirical findings and limitations (Section 6, p. 25).
|
|
Are all figures and tables clear and well-presented? |
Must be improved |
Figures were simplified and captions expanded for clarity (Figures 3–9, pp. 11–22). |
|
3. Point-by-point response to Comments and Suggestions for Authors |
||
|
Comments 1: [Novelty and Contribution.] “The contribution is incremental rather than conceptually novel. The paper should tone down claims of originality and emphasize the value of comprehensive comparative evaluation rather than methodological innovation.”
|
||
|
Response 1: We fully agree. The manuscript has been revised to clearly position the study as a systematic benchmarking and applied preprocessing evaluation, rather than as a novel algorithmic contribution. Claims of methodological novelty were toned down throughout, and emphasis was shifted toward the breadth, rigor, and reproducibility of the comparative evaluation across datasets and model architectures.
Revisions:
|
||
|
Comments 2: [Methodological Clarity] “The methodology is described in excessive detail. Justification of percentile thresholds and computational overhead is missing.”
|
||
|
Response 2: We agree. Overly detailed tutorial-style explanations were condensed, and explicit methodological justifications were added. In particular, we clarified the rationale for percentile thresholds based on anatomical consistency and empirical validation across datasets. We also added a clear discussion of the computational overhead introduced by the adaptive pipeline.
|
||
|
Revisions:
· Abstract (p. 1): Reports overall overhead (+6.3%, 5.2 ms per batch). · Discussion (Section 5, p. 23-24): Provides interpretive context and practical implications.
Comment 3: [Experimental Design and Ablation] “Most improvements are small and sometimes inconsistent. A dedicated ablation of cropping only vs histogram only would help isolate contributions.”
Response 3:
Revisions:
Comment 4: [Interpretation of Results and Bias] “The discussion is descriptive rather than analytical. Bias (e.g., pediatric vs adult) and qualitative visualization are missing.”
Response 4:
Revisions:
Comment 5: [Writing Quality, Structure, and Related Work] “Sections are too long and repetitive; related work is unfocused and misses transformers and foundation models.”
Response 5:
Revisions:
|
||
|
4. Response to Comments on the Quality of English Language |
||
|
Reviewer comment: “The English is fine and does not require any improvement.”
Response: |
||
|
|
||
|
5. Additional clarifications |
||
|
||
Reviewer 2 Report (Previous Reviewer 1)
Comments and Suggestions for Authors
The authors significantly improved the work, I think it is ready.
Author Response
We sincerely thank for the thoughtful evaluation and supportive feedback. Your positive assessment is greatly appreciated and has been truly encouraging for our research team. We are grateful for your time and expertise in reviewing our work.
Round 2
Reviewer 1 Report (Previous Reviewer 3)
Comments and Suggestions for Authors
The revised manuscript demonstrates a substantial and well-documented effort to address the comments raised in previous review rounds. The authors have clearly repositioned the study as a systematic and empirical benchmarking analysis of normalization strategies for chest X-ray classification, rather than as a proposal of a novel algorithmic method. This clarification significantly improves the conceptual integrity of the work and aligns the stated contributions with what is actually delivered experimentally. The revised abstract, introduction, and contribution statements are now appropriately scoped and avoid overstating novelty, which was a central concern in earlier rounds.
From a methodological standpoint, the experimental design is now clearly articulated and well controlled. The use of four public chest X-ray datasets with standardized sampling, three convolutional architectures of varying complexity, and identical training protocols enables a fair comparison of normalization strategies under cross-dataset conditions. The detailed description of the adaptive normalization pipeline, including percentile-based ROI cropping and histogram standardization, is sufficiently transparent and reproducible. Importantly, the authors explicitly acknowledge that the proposed pipeline is motivated by prior work and should be interpreted as a practical preprocessing design choice rather than a new theoretical contribution. This clarification is appropriate and resolves earlier ambiguity.
The experimental results are extensive and generally well interpreted. The authors provide consistent evidence that adaptive normalization improves or maintains performance on datasets with relatively stable intensity distributions, while offering limited benefit on MIMIC-CXR due to its substantial acquisition heterogeneity. This dataset-dependent behavior is now explicitly discussed rather than glossed over, which strengthens the credibility of the conclusions. The inclusion of statistical significance testing using Friedman–Nemenyi and Wilcoxon signed-rank tests further improves the rigor of the analysis and supports the reported performance differences beyond descriptive trends alone.
The ablation study isolating the effects of ROI cropping and histogram standardization is a particularly valuable addition in this revision. It demonstrates that histogram standardization provides the dominant performance gain, while cropping offers complementary improvements, especially for lightweight architectures such as MobileNetV2. This analysis helps disentangle the contributions of individual preprocessing components and provides practical guidance for future system design. The discussion appropriately emphasizes that normalization effectiveness depends on dataset characteristics and model architecture, rather than implying a universally optimal solution.
Despite these strengths, the manuscript remains quite long, and some sections—particularly in the methodology and results—contain a level of detail that could be moderately reduced without loss of scientific clarity. Additionally, while the discussion of clinical implications is thoughtful, it remains largely conceptual; no clinician-in-the-loop evaluation or external clinical validation is attempted. These limitations are, however, explicitly acknowledged by the authors and do not undermine the core empirical findings.
Overall, this revised version satisfactorily addresses the major concerns raised in previous review rounds. The manuscript now presents a clear, honest, and statistically grounded study on the role of normalization in cross-dataset chest X-ray classification. While the contribution is primarily empirical and applied rather than methodological, it is well executed and relevant to the medical imaging community.
Recommendation: Accept after minor editorial refinement, mainly focused on further reducing redundancy and tightening the presentation.
Author Response
|
Response to Reviewers’ Comments Manuscript Title: Adaptive Normalization Enhances the Generalization of Deep Learning Model in Chest X-ray Classification
We sincerely thank the reviewer for the thorough and constructive evaluation of our manuscript. We greatly appreciate the reviewer’s recognition of the substantial revisions made in this round and the overall positive assessment of the manuscript. In response to the reviewer’s comments, we have performed minor editorial refinements aimed at further tightening the presentation, reducing redundancy, and improving clarity, while preserving methodological transparency and reproducibility. All revisions have been incorporated into the revised manuscript.
|
||
|
1. Summary |
|
|
|
Thank you very much for taking the time to review this manuscript. Please find the detailed responses below, with corresponding revisions highlighted using track changes in the resubmitted manuscript. Following the reviewer’s suggestions, the manuscript has been refined to:
Redundant explanations—particularly in Sections 3 and 4, were reduced, and figure and table descriptions were tightened for conciseness, in line with the reviewer’s recommendation for minor editorial refinement.
|
||
|
2. Questions for General Evaluation |
Reviewer’s Evaluation |
Response and Revisions |
|
Does the introduction provide sufficient background and include all relevant references? |
Can be improved |
The Introduction was refined to emphasize the specific gap in normalization benchmarking under cross-dataset conditions and to clearly frame the contribution as an empirical evaluation (Section 1). |
|
Is the research design appropriate?
|
Can be improved |
The experimental design was clarified by explicitly highlighting controlled dataset sampling, identical training protocols, and multi-seed evaluation (Section 3.4). |
|
Are the methods adequately described? |
Can be improved |
Method descriptions were streamlined, and explicit justification for ROI percentile thresholds and histogram standardization was added (Sections 3.2–3.4; Algorithm 1). |
|
Are the results clearly presented? |
Can be improved |
Results were reorganized using composite figures and tables, and a qualitative error analysis was added (Section 4; Figures 6–9; Tables 5–8). |
|
Are the conclusions supported by the results? |
Can be improved |
Conclusions were explicitly aligned with empirical findings and dataset-dependent limitations (Section 6). |
|
Are all figures and tables clear and well-presented? |
Can be improved |
Figure captions were expanded and tables were reformatted for clarity and consistency (Figures 3–9; Tables 5–8). |
|
3. Point-by-point response to Comments and Suggestions for Authors |
||
|
Comments 1: [Novelty and Contribution.] “The contribution is incremental rather than conceptually novel.”
|
||
|
Response 1: We fully agree. The manuscript was revised to clearly present the work as a systematic benchmarking and applied preprocessing study, rather than a novel methodological contribution. Claims of algorithmic novelty were removed or toned down throughout.
Revisions: · Abstract: Explicitly states the empirical and applied nature of the study. · Introduction: Reframed to emphasize benchmarking and cross-dataset robustness. · Conclusions: Reinforce the incremental, practice-oriented contribution.
|
||
|
Comments 2: [Methodological Clarity] “Justification of percentile thresholds and computational overhead is missing.”
|
||
|
Response 2: We added an explicit justification for the ROI percentile thresholds based on anatomical consistency considerations and exploratory checks across all four datasets. We also quantified the computational overhead of the adaptive pipeline and reported it in the Abstract and discussed it in Section 5.
|
||
|
Revisions: · ROI percentile justification: Section 3.2.3. · Computational overhead: Abstract and Discussion (Section 5).
Comment 3: [Ablation Study] “A dedicated ablation of cropping vs. histogram standardization is needed.”
Response 3:
Revisions:
Comment 4: [Interpretation, Bias, and Visualization] “Bias (e.g., pediatric vs. adult) and qualitative visualization are missing.”
Response 4: The Discussion was expanded to address dataset-dependent behavior (notably MIMIC-CXR), and to discuss a potential pediatric anatomical bias when applying adult-derived cropping parameters. We explicitly acknowledge that no dedicated pediatric subgroup analysis was performed in this work. In addition, a brief qualitative illustration of misclassified cases was included (Figure 9) to support interpretation.
Revisions:
Comment 5: [Length and Related Work] “The manuscript remains quite long, and some sections—particularly in the methodology and results—contain a level of detail that could be moderately reduced without loss of scientific clarity. Minor editorial refinement is recommended to further tighten the presentation.”
Response 5: We thank the reviewer for this helpful suggestion. In response, we carefully revised the manuscript to reduce redundancy and improve conciseness, while preserving methodological transparency and reproducibility. Specifically, overly detailed or repetitive explanations in the methodology and results sections were condensed, figure and table descriptions were tightened, and the overall presentation was streamlined. These revisions aim to improve readability without altering the experimental design, analysis, or conclusions of the study.
Revisions: · Reduced redundancy and condensed methodological and results descriptions: Sections 3–4. · Tightened figure and table captions for clarity and conciseness. · Minor editorial refinements applied throughout the manuscript.
|
||
|
4. Response to Comments on the Quality of English Language |
||
|
Reviewer comment: “The English is fine and does not require any improvement.”
Response: |
||
|
|
||
|
5. Final Remarks |
||
|
||
This manuscript is a resubmission of an earlier submission. The following is a list of the peer review reports and author responses from that submission.
Round 1
Reviewer 1 Report
Comments and Suggestions for Authors
The authors compared the image normalization methods across X-ray datasets, it is fundamental topic which could direct the research path, but there are some drawbacks should be addressed to improve the quality.
1) Did you consider the impact of data augmentation in your experiments?
2) could you please provide the hyperparameters in you experiments?
3) You could provide detailed metrics, such as sensitivity, specificity, and the significance.
4) Fig. 1 and Fig. 2 are not complete.
Reviewer 2 Report
Comments and Suggestions for Authors
1. It is needed a revision about the English language.
2. the manuscript is not considered well-structured. It is hard to follow this manuscript.
3. The quality of the article is very poor. See Page 6 and 7, the guidelines of Journal Submission are included. Do you check your manuscript? As a reviewer, I feel angry.
4. Figure 4 and 5 show the codes. It is a manuscript, not a report.
5. The novelty of this manuscript is not considered good.
I recommend the rejection of this manuscript.
Reviewer 3 Report
Comments and Suggestions for Authors
This manuscript investigates how different image normalization strategies affect the generalization capability of deep learning models for chest X-ray (CXR) classification. The authors compare three preprocessing approaches—scaling, Z-score, and a proposed adaptive normalization (combining percentile-based cropping and histogram standardization)—across four benchmark datasets (ChestX-ray14, CheXpert, MIMIC-CXR, and Chest X-ray Pneumonia) and three CNN architectures (custom lightweight CNN, EfficientNet-B0, MobileNetV2). The study concludes that adaptive normalization improves validation stability, accuracy, and F1-score, particularly under domain shifts, with statistical significance confirmed via Friedman and Wilcoxon tests.
The paper is technically sound, methodically structured, and highly relevant to J. Imaging. Its experimental breadth and emphasis on preprocessing as a generalization factor are timely and well justified. However, the manuscript would benefit from minor revisions in clarity, figure presentation, and discussion focus, particularly to strengthen its clinical and computational interpretations.
1. Abstract and Keywords
- Suggest simplifying minor redundancy (e.g., “scaling, Z-score, and an adaptive approach” can be condensed).
- Keep as is with light editorial trimming for brevity.
2. Introduction
- Clarify the rationale for the adaptive normalization method earlier—perhaps include a motivating schematic or figure comparing traditional vs. adaptive pipelines.
- Avoid small redundancies (e.g., “due to being diverse in demographics…” → “owing to diverse demographics…”).
3. Methodology
- Provide more detail on ROI cropping thresholds—why 5–95% for x-axis and 15–95% for y-axis? Were these empirically optimized?
- Clarify whether the target mean/std (μ_target, σ_target) were derived from any reference dataset or computed heuristically.
- Figure captions should be slightly more self-contained (e.g., “CDF-guided cropping for ROI localization” rather than “Horizontal and vertical grayscale sum distributions”).
- Training code snippet (Fig. 5) is not essential for journal readability—can be summarized as pseudocode or moved to supplementary material.
4. Experimental Results
- Figures 7–18 are too many and repetitive; condense results by grouping related metrics (e.g., one composite figure per dataset or model).
- Accuracy and F1 differences among methods are sometimes small (< 2%); emphasize statistical significance rather than raw percentages.
- The adaptive normalization improvements on MIMIC-CXR are less pronounced—authors should discuss potential dataset-specific reasons (scanner heterogeneity, label noise).
- Include at least one qualitative visualization (e.g., cropped vs. original CXR or Grad-CAM outputs) to demonstrate interpretability.
5. Discussion
- Portions are repetitive with the conclusion and could be condensed.
- The paragraph on computational overhead (lines 705–710) is important—consider quantifying this overhead in seconds/epoch or % slowdown.
- Condense to highlight the conceptual significance (normalization as a “clinical enabler”) and minimize repetition.
6. Writing Quality and Presentation
- A few syntactic corrections and smoother transitions are needed (especially in Sections 2–3).
- Figures and tables are consistent with MDPI standards; ensure high-resolution versions for publication.
- Minor language polishing before final submission.
7. References
- Minor consistency issue: ensure all references follow MDPI format (DOI in lowercase, correct use of italics for journal titles).